# Pharmacokinetics and Tissue Distribution of Enavogliflozin in Mice and Rats

**DOI:** 10.3390/pharmaceutics14061210

**Published:** 2022-06-07

**Authors:** Minyeong Pang, So Yeon Jeon, Min-Koo Choi, Ji-Hyeon Jeon, Hye-Young Ji, Ji-Soo Choi, Im-Sook Song

**Affiliations:** 1College of Pharmacy, Dankook University, Cheonan-si 31116, Korea; whatsupmy@naver.com (M.P.); ojsw97@naver.com (S.Y.J.); minkoochoi@dankook.ac.kr (M.-K.C.); 2BK21 FOUR Community-Based Intelligent Novel Drug Discovery Education Unit, Vessel-Organ Interaction Research Center (VOICE), Research Institute of Pharmaceutical Sciences, College of Pharmacy, Kyungpook National University, Daegu 41566, Korea; kei7016@naver.com; 3Life Science Institute, Daewoong Pharmaceutical, Yongin 17028, Korea; hy-chi138@daewoong.co.kr (H.-Y.J.); jschoi172@daewoong.co.kr (J.-S.C.)

**Keywords:** sodium-glucose cotransporter 2 inhibitors, enavogliflozin, pharmacokinetics, kidney distribution

## Abstract

This study investigated the pharmacokinetics and tissue distribution of enavogliflozin, a novel sodium-glucose cotransporter 2 inhibitor that is currently in phase three clinical trials. Enavogliflozin showed dose-proportional pharmacokinetics following intravenous and oral administration (doses of 0.3, 1, and 3 mg/kg) in both mice and rats. Oral bioavailability was 84.5–97.2% for mice and 56.3–62.1% for rats. Recovery of enavogliflozin as parent form from feces and urine was 39.3 ± 3.5% and 6.6 ± 0.7%, respectively, 72 h after its intravenous injection (1 mg/kg), suggesting higher biliary than urinary excretion in mice. Major biliary excretion was also suggested for rats, with 15.9 ± 5.9% in fecal recovery and 0.7 ± 0.2% in urinary recovery for 72 h, following intravenous injection (1 mg/kg). Enavogliflozin was highly distributed to the kidney, which was evidenced by the AUC ratio of kidney to plasma (i.e., 41.9 ± 7.7 in mice following its oral administration of 1 mg/kg) and showed slow elimination from the kidney (i.e., T_1/2_ of 29 h). It was also substantially distributed to the liver, stomach, and small and large intestine. In addition, the tissue distribution of enavogliflozin after single oral administration was not significantly altered by repeated oral administration for 7 days or 14 days. Overall, enavogliflozin displayed linear pharmacokinetics following intravenous and oral administration, significant kidney distribution, and favorable biliary excretion, but it was not accumulated in the plasma and major distributed tissues, following repeated oral administration for 2 weeks. These features may be beneficial for drug efficacy. However, species differences between rats and mice in metabolism and oral bioavailability should be considered as drug development continues.

## 1. Introduction

Sodium-glucose cotransporter 2 (SGLT2) is abundantly expressed in the S1 segment of the proximal kidney and plays a major role in the reabsorption of filtered glucose [1]. Based on this mechanism, SGLT2 inhibitors have been developed as antidiabetic drugs that achieve glycemic control by inhibiting renal tubular glucose reabsorption [2] and reducing the risk of hypoglycemia [3]. Additionally, SGLT2 inhibitors show beneficial effects on cardiovascular risk and nephrotoxicity [4,5,6,7,8,9].

Several SGLT2 inhibitors, such as canagliflozin, dapagliflozin, empagliflozin, and ipragliflozin etc., are approved for the treatment of type 2 diabetes [3,10]. These SGLT2 inhibitors show higher inhibitory potency for SGLT2 than for SGLT1 (more than 400-fold) and half-maximal inhibitory concentrations (IC_50_) for SGLT2 are in the low nanomolar range (less than 10 nM) [11]. Canagliflozin reduced HbA1c by 0.97% in untreated type 2 diabetic (T2DM) patients [12]. A meta-analysis indicated that canagliflozin showed similar hypoglycemic control compared to metformin monotherapy and sitagliptin therapy [13,14]. Such results were also reported for other SGLT2 inhibitors. Dapagliflozin monotherapy (2.5–10 mg, once daily for 24 weeks) induced a significant reduction in HbA1c values (in the range of 0.67–0.84%) in untreated T2DM patients [15]. Another meta-analysis reported that dapagliflozin, with conventional anti-diabetes medication, such as metformin, sulfonylurea, sitagliptin, and thiazolidinediones, reduced mean HbA1c by 0.54% compared with conventional monotherapy [4,16,17]. HbA1c was reduced by 0.74–0.85% in empagliflozin-treated patients and a significant reduction of HbA1c was achieved by empagliflozin add-on therapy compared with metformin monotherapy or pioglitazone, with, or without, metformin treatment [18,19]. 

SGLT2 inhibitors draw special interest because of their cardioprotective activity and their ability to reduce risks of kidney failure [1]. Another recent meta-analysis of cardiovascular safety with empagliflozin (EMPA-RGE OUTCOME), canagliflozin (CANVAS Program), and dapagliflozin (DECLARE-TIMI) indicated that SGLT2 inhibitors significantly reduced major cardiac events, such as myocardial infarction, stroke, and cardiovascular death by 11% [11]. In addition, sub-analysis from the EMPA-RGE OUTCOME results showed that empagliflozin treatment in diabetic patients with cardiovascular disease reduced the incidence or worsening of nephrotoxicity [9]. Recently, Tahara et al. [10] reported prolonged glucose-lowering efficacy of SGLT2 inhibitors closely correlated with drug distribution, retention in the kidney, and elimination half-life. Notably, SGLT2 is the primary contributor, contributing about 90% to renal glucose reabsorption [3]. Thus, kidney distribution and elimination profiles of SGLT2 inhibitors and potent SGLT2 inhibition are important for efficacy. 

Enavogliflozin [DWP16001; (2S,3R,4R,5S,6R)-2-(7-chloro-6-(4-cyclopropylbenzyl) -2,3-dihydrobenzofuran-4-yl)-6-(hydroxymethyl)tetrahydro-2H-pyran-3,4,5-triol] (Figure 1), a selective SGLT2 inhibitor, is under development by Daewoong Pharmaceutical Co. Ltd. (Seoul, Korea) and is currently in phase three clinical trials (Registration No. NCT04654390 at www.clinicaltrials.gov accessed on 18 April 2022). Enavogliflozin selectively inhibits SGLT2 showing a 667-fold difference in IC_50_ values for SGLT2 versus SGLT1. IC_50_ values for SGLT2 and SGLT1 were 0.8 ± 0.3 nM and 549.3 ± 139.6 nM, respectively. It reversibly and competitively inhibited SGLT2, but it showed restrained recovery of the SGLT2 activity after the removal of enavogliflozin [20]. Enavogliflozin showed significantly higher kidney distribution compared to dapagliflozin and ipragliflozin. The ratio of area under the concentration curve (AUC) in kidney to AUC in plasma was 85.0 ± 16.1, 64.6 ± 31.8, and 38.4 ± 5.3, respectively [20], reflecting the potential therapeutic efficacy of the drug. Hwang et al. reported a dose-dependent increase in urinary glucose excretion after a single oral administration of enavogliflozin in a dose range of 0.2 to 5.0 mg in healthy male volunteers (Registration No. NCT03364985 at www.clinicaltrials.gov accessed on 18 April 2022) [21], suggesting a dose-dependent inhibition of glucose reabsorption as the result of SGLT2 inhibition. Thus, this study aimed to investigate the dose-dependency in the pharmacokinetics of enavogliflozin in mice and rats and to investigate the tissue distribution and excretion of enavogliflozin in mice. In a study of comparative pharmacokinetic, pharmacodynamic and pharmacological effects of various SGLT2 inhibitors in rats, oral dose ranges of 0.3–3 mg/kg for ipragliflozin and dapagliflozin and 1–10 mg/kg for tofogliflozin, canagliflozin, empagliflozin, and luseogliflozin were used as the effective dose ranges [10]. Considering the effective dose of various SGLT2 inhibitors and IC_50_ values of enavogliflozin for SGLT2 and SGLT1 [10,20], we selected 0.3, 1, and 3 mg/kg as oral and intravenous doses of enavogliflozin in this study. 

## 2. Materials and Methods

### 2.1. Materials

Enavogliflozin (Lot No. E2016-085-27-2) (Figure 1) and d4-enavogliflozin as an internal standard (IS), were obtained from Daewoong Pharmaceutical Co. Ltd. (Seoul, Korea). Methyl tert-butyl ether (MTBE) was purchased from J.T. Baker (Phillipsburg, NJ, USA). Methanol was purchased from TEDIA (Fairfield, OH, USA). All other chemicals and solvents were reagent or analytical grade. 

### 2.2. Pharmacokinetic Study

Male Institute of Cancer Research (ICR) mice (7 weeks old, 27–33 g) and male Sprague Dawley (SD) rats (7 weeks old, 230–250 g) were purchased from Samtako Co. (Osan, Kyunggido, Korea). The animals were acclimatized for one week in an animal facility at the College of Pharmacy, Kyungpook National University. Food and water were available ad libitum.

#### 2.2.1. Pharmacokinetic Study

Fifty-four ICR mice were randomly divided into nine groups (n = 6 per group; Table 1) and intravenously administered an enavogliflozin solution at doses of 0.3, 1, and 3 mg/kg. The drug was dissolved in a mixture of 10% DMSO and 90% saline and injected via the tail vein. Before blood sampling, mice were anesthetized for 5 min using 2% isoflurane in a vaporizer with an oxygen flow of 0.8 L/min. Blood sampling used a sparse sampling method via the right or left retro-orbital vein using heparinized capillary tubes (Heinz Herenz, Hamburg, Germany). Final blood was collected from the abdominal artery using a heparin-treated 1 mL syringe (Jung Lim Co. Ltd., Choong-Buk, Korea) under isoflurane anesthesia (Table 1). Blood samples were collected at 0, 0.083, 0.25, 0.5, 1, 2, 4, 8, and 24 h and centrifuged at 12,000× g for 1 min to separate plasma. An aliquot (30 µL) of each plasma sample was stored at −80 °C until enavogliflozin analysis. 

Mice were fasted with water ad libitum for at least 12 h before oral administration with enavogliflozin. Fifty-four mice were randomly divided into nine groups (n = 6 per group, Table 1) and were administered an enavogliflozin solution at doses of 0.3, 1, and 3 mg/kg via oral gavage. The drug was dissolved in a mixture of 10% DMSO and 90% saline. Blood samples were collected at 0, 0.083, 0.25, 0.5, 1, 2, 4, 8, and 24 h with the same procedures described above and provided in Table 1. 

Three mice received enavogliflozin (1 mg/kg) intravenously and were returned to metabolic cages with food and water ad libitum and urine and feces samples were collected every 24 h for 72 h. Urine and feces were weighed, and 30 µL aliquots of urine and 50 µL aliquots of 10% feces homogenates were stored at −80 °C until enavogliflozin analysis. Four mice received enavogliflozin (1 mg/kg) by oral gavage and were returned to their metabolic cages to collect urine and feces samples every 24 h for 72 h with the same protocols described above. 

Twelve rats were randomly divided into three groups (n = 4 per each group) and injected with an enavogliflozin solution at doses of 0.3, 1, and 3 mg/kg. The drug was dissolved in a mixture of 10% DMSO and 90% saline and administered intravenously via the tail vein. Before blood sampling, rats were anesthetized using 2% isoflurane in a vaporizer with an oxygen flow of 0.8 L/min, for 5 min. Blood samples (approximately 100 µL) were collected at 0, 0.05, 0.33, 0.67, 1, 2, 4, 6, and 8 h via the jugular vein under isoflurane anesthesia using a heparin-treated 1 mL syringe (Jung Lim Co. Ltd., Jincheon, Korea). Samples were centrifuged at 12,000× *g* for 1 min to separate the plasma. An aliquot (30 µL) of each plasma sample was stored at −80 °C until analysis. 

Rats were fasted with water ad libitum for at least 12 h before the oral administration of enavogliflozin. Eighteen rats were randomly divided into three groups (n = 6 per each group) and administered the drug dissolved in a mixture of 10% DMSO and 90% saline at doses of 0.3, 1, and 3 mg/kg by oral gavage. Blood samples were collected at 0, 0.083, 0.25, 0.5, 1, 2, 4, 6, 8, and 24 h with the same procedures described above. 

Three rats received enavogliflozin (1 mg/kg) intravenously and were returned to metabolic cages with food and water ad libitum and urine and feces samples were collected every 24 h for 72 h. Four rats received enavogliflozin (1 mg/kg) by oral gavage and were returned to their metabolic cages to collect urine and feces samples every 24 h for 72 h with the same protocols described above.

#### 2.2.2. Tissue Distribution Study

Thirty-six ICR mice were fasted with water ad libitum for at least 12 h before oral administration of enavogliflozin and randomly divided into six groups (n = 6 per each sampling time point) and administered with an enavogliflozin solution at a dose of 1 mg/kg via oral gavage. Blood samples (approximately 0.2 mL) were collected at 0.5, 1, 2, 4, 8, 24 h via the abdominal artery. Subsequently, whole tissues, including stomach, small intestine, large intestine, liver, kidney, brain, heart, lung, spleen, and testis, were isolated. Blood samples were centrifuged at 12,000× *g* for 1 min to separate plasma. Tissue samples were minced thoroughly and homogenized with four volumes of saline using a tissue grinder. An aliquot (30 µL) of plasma and aliquots (50 µL) of tissue homogenate samples were stored at −80 °C until analysis. 

Thirty-five ICR mice were randomly divided into five groups (n = 7 per sampling time point) and administered with an enavogliflozin solution (1 mg/kg, once daily) for 14 days via oral gavage. Another thirty-five ICR mice were randomly divided into five groups (n = 7 per sampling time point) and administered with an enavogliflozin solution (1 mg/kg, once daily) for 7 days via oral gavage. The other thirty-five ICR mice were randomly divided into five groups (n = 7 per sampling time point) and administered with an enavogliflozin solution at a dose of 1 mg/kg via oral gavage. Blood samples (approximately 0.2 mL) were collected at 1, 2, 8, 24, and 48 h via the abdominal artery. Subsequently, whole tissues, including kidney, liver, small intestine, and large intestine, were isolated. An aliquot (30 µL) of plasma and aliquots (50 µL) of tissue homogenate samples were prepared and stored with the same procedures described above. 

### 2.3. LC-MS/MS Analysis of Enavogliflozin

Concentrations of enavogliflozin in plasma and tissue homogenate samples were analyzed using an Agilent 6430 triple quadrupole liquid chromatography-mass spectrometry (LC–MS/MS) system (Agilent, Wilmington, DE, USA) following a previously published method [20].

Aliquots of plasma or urine (30 µL each) and tissue homogenate (50 µL each) were added to 100 µL of aqueous solution of d4-enavogliflozin (IS, 20 ng/mL), and vigorously mixed with 500 µL MTBE for 15 min. After centrifugation at 16,000× *g* for 5 min, samples were kept for 1 h, at a temperature of −80 °C to freeze the aqueous layer freeze. The organic upper layer was then transferred to a clean tube and evaporated to dryness under a gentle stream of nitrogen. The dried extract was reconstituted in 150 µL of mobile phase, and a 3 µL aliquot was injected into the LC–MS/MS system. Enavogliflozin was separated on a Synergi Polar RP column (2.0 × 150 mm, 4 µm particle size; Phenomenex, Torrence, CA, USA) using a isocratic mobile phase consisting of water (15%) and methanol (85%) containing 0.1% formic acid at a flow rate of 0.25 mL/min. 

Quantification of the analyte peak at *m*/*z* 464 → 131 for enavogliflozin (T_R_ (retention time) 2.8 min), and *m*/*z* 468 → 135 for d4-enavogliflozin (T_R_ 2.8 min) used positive ionization mode with a collision energy of 25 eV. The calibration standards of enavogliflozin in the plasma and tissue homogenates were linear in the range of 5–3000 ng/mL. The inter-day and intra-day precision and accuracy were within 15% for respective quality control samples (5, 15, 250, and 2000 ng/mL). Extraction recovery and matrix effect were in the range of 80.7–89.0% and 98.40–108.2%, respectively. 

### 2.4. Statistics

The data were expressed as the means ± standard deviation for the groups. Pharmacokinetic parameters, such as the area under the plasma concentration–time curve during the period of observation (AUC_last_), AUC to infinite time (AUC_∞_), clearance (CL), and volume of distribution at steady-state (V_d,ss_), the terminal half-life (t_1/2_), and mean residence time (MRT) were calculated using non-compartment analysis with WinNonlin software (version 5.1; Pharsights, Cary, NC, USA). The AUC ratios were calculated by dividing the AUC_last_ of enavogliflozin in the tissue samples by the plasma AUC_last_ values of enavogliflozin [15].

The normal distribution of the data was assessed using the Shapiro-Wilk test for normality and comparisons of the pharmacokinetic parameters (i.e., AUC_∞_/D, C_o_/D, CL, V_d,ss_, T_1/2_, MRT in both mice and rats following intravenous administration of enavogliflozin; AUC_∞_/D, C_max_/D, T_max_, T_1/2_, MRT in both mice and rats following oral administration of enavogliflozin; AUC and AUC ratio in mice following single or repeated oral administration of enavogliflozin) were made for three groups using the non-parametric Kruskal-Wallis test because of the small number of the sample size. SPSS for Windows software (version 25.0; IBM Corp., Armonk, NY, USA) was used and a difference was considered significant at *p* < 0.05.

## 3. Results

### 3.1. Pharmacokinetics of Enavogliflozin in Mice

AUC_∞_ and C_o_ of enavogliflozin in ICR mouse plasma increased with increasing intravenous doses of 0.3, 1, and 3 mg/kg following intravenous injection (Figure 2A, Table 2). The normality test using the Shapiro-Wilk method indicated that the pharmacokinetic parameters (i.e., Dose normalized AUC (AUC_∞_/D), and dose normalized initial concentration (C_o_/D), CL, V_d,ss_, T_1/2_, and MRT] showed normal distribution (Appendix A). The Kruskal-Wallis test for these kinetic parameters recognized no significant differences in these pharmacokinetic parameters (Table 2). Thus, enavogliflozin displayed linear kinetics in an intravenous dose range of 0.3–3 mg/kg. 

The plasma concentration-time profile of enavogliflozin in mice following oral administration of enavogliflozin is shown in Figure 2B and the respective pharmacokinetic parameters are summarized in Table 2. The pharmacokinetic parameters of orally administered enavogliflozin showed normal distribution using the Shapiro-Wilk test (Appendix A) and the dose correlation among the pharmacokinetic parameters of the three dose groups were tested using the Kruskal-Wallis test. Dose normalized maximum plasma concentration (C_max_/D), AUC_∞_/D, and time to reach C_max_ (T_max_) obtained after administration of doses of 0.3, 1, and 3 mg/kg, showed no significant differences in the three different dosing groups (Figure 2B and Table 2; *p* > 0.05 using Kruskal-Wallis test). Thus, enavogliflozin pharmacokinetic parameters obtained after its oral administration showed no significant differences in the oral dose range of 0.3–3 mg/kg. The oral bioavailability of enavogliflozin was 97.2%, 84.5%, and 93.7% for doses of 0.3, 1, and 3 mg/kg, respectively (Table 2).

### 3.2. Pharmacokinetics of Enavogliflozin in Rats

Plasma concentration-time profiles of enavogliflozin in rats following intravenous injection of enavogliflozin were similar to the results found in mice (Figure 3A and Table 3). No significant differences were observed for C_o_/D, AUC_∞_/D, CL, and V_d__,ss_ (Table 3; *p* > 0.05 using Kruskal-Wallis test), and normal distribution of these parameters was confirmed by the Shapiro-Wilk test (Appendix A). Thus, enavogliflozin also showed linear kinetics in rats in the intravenous dose range of 0.3–3 mg/kg. This was evidenced by a dose-proportional increase of AUC values of enavogliflozin in both rats and mice (Figure 4A,B). The T_1/2_ and MRT values obtained from 0.3, 1, and 3 mg/kg groups were significantly different. However, the plasma concentrations of enavogliflozin at 24 h following intravenous administrations of 0.3, 1, and 3 mg/kg groups were below the detection limit and resulted in incomplete elimination phase to estimate T_1/2_ and MRT (Table 3).

Similarly, plasma concentration-time profiles of enavogliflozin in rats following oral administration of enavogliflozin were also similar to results from mice (Figure 3B and Table 3). Normal distribution of these parameters was confirmed by the Shapiro-Wilk test (Appendix A) but no significant differences in C_max_/D, AUC/D, and T_max_ were observed (Table 3; *p* > 0.05 using Kruskal-Wallis test). Thus, enavogliflozin also showed linear kinetics in rats in an oral dose range of 0.3–3 mg/kg. Similarly, AUC values of enavogliflozin increased dose proportionally in both rats and mice (Figure 4C,D). In addition, the oral bioavailability of enavogliflozin in rats at doses of 0.3, 1, and 3 mg/kg was 62.1%, 58.9%, and 56.3%, respectively (Table 3). These values were lower than values obtained from mice treated with the same doses. 

### 3.3. Recovery of Enavogliflozin in Mice and Rats

#### 3.3.1. Recovery of Enavogliflozin in Mice

Recovery of enavogliflozin was assessed from the urine and the feces samples collected over 72 h for mice and rats. The amount of enavogliflozin recovered in feces for 24 h and for 72 h after dosing was 36.3% and 39.3%, respectively (Table 4). The amount of enavogliflozin recovered in urine for 24 h and for 72 h was 6.3% and 6.6%, respectively (Table 4). Thus, most enavogliflozin was eliminated from the body within 24 h. Recovery of enavogliflozin in feces was about 6-fold greater than in urine and enavogliflozin seemed to be excreted mainly via the biliary route in mice. However, total recovery of enavogliflozin was about 46%, suggesting that enavogliflozin was metabolized prior to elimination. 

After oral administration (1 mg/kg) of enavogliflozin in mice, most enavogliflozin was again recovered within 24 h and recovery from feces was much greater than from urine (Table 4), consistent with recovery after intravenous administration. However, recovery from feces after 72 h was 51.4%. Considering the BA of enavogliflozin in mice was 84.5%, unabsorbed fraction could have contributed to the increased fecal recovery. 

#### 3.3.2. Recovery of Enavogliflozin in Rats

Recovery of enavogliflozin in urine and feces was assessed in rats over 72 h. The amount recovered in feces and urine was 15.9% and 0.7%, respectively, after IV dose, and most enavogliflozin was recovered within 24 h (Table 5). The recovery of enavogliflozin in feces was about 22.7-fold greater than in urine and enavogliflozin seemed to be excreted mainly via the biliary route, similar to the case of mice. However, total recovery of enavogliflozin was about 16.6%, suggesting greater metabolism before elimination in rats than in mice. These data demonstrated that elimination and extent of metabolism showed species differences. 

Most enavogliflozin was also recovered within 24 h after oral administration of enavogliflozin and recovery from feces was much greater than from urine (Table 5), consistent with recovery after intravenous administration. However, recovery from feces after 72 h was 45.5%. Again, this finding might be attributed to unabsorbed drug, considering the BA of enavogliflozin (58.9%).

### 3.4. Tissue Distribution of Enavogliflozin in Mice

#### 3.4.1. Single Oral Administration of Enavogliflozin in Mice

Concentrations of enavogliflozin in ten tissues were assessed; the drug was not detected in brain tissue and enavogliflozin concentration and its elimination in nine tissues varied (Figure 5). Elimination constants K and T_1/2_ of enavogliflozin in various tissues are summarized in Table 6. The kidney showed a prolonged half-life and the highest enavogliflozin concentrations among the ten tissues in mice. Enavogliflozin in the stomach, small intestine, large intestine, liver was higher than in plasma and, consequently, showed a greater AUC ratio (higher than 5-fold) (Table 6). Enavogliflozin concentrations in the heart and lung were similar to plasma concentrations and those in testis and spleen were lower than plasma (Figure 5). The half-lives in the lung and testis were longer than in plasma but elimination half-lives in other tissues were similar to the half-life in plasma (Table 6). 

Tissue distribution (AUC ratios) of enavogliflozin in the kidney, stomach, small intestine, large intestine, liver, heart, and lung were higher than unity, whereas distribution in the brain, spleen, and testis was lower than unity (Table 6). The order of AUC ratios was kidney > stomach, small intestine, large intestine > liver > lung, heart > testis, spleen > brain.

#### 3.4.2. Tissue Distribution of Enavogliflozin following Repeated Oral Administration of Enavogliflozin in Mice

To investigate the effect of repeated doses of enavogliflozin on the pharmacokinetics and tissue distribution of enavogliflozin, this study measured the enavogliflozin concentrations in the plasma and major distributed tissues, such as the kidney, liver, small intestine, and large intestine for 48 h following single or repeated oral doses of enavogliflozin for 7 or 14 days (Figure 6) and AUC values and AUC ratios are shown in Table 7. Normal distribution of AUC values and AUC ratios from different tissues and treatment groups were assessed by the Shapiro-Wilk test (Appendix A). The plasma concentration-time profile of enavogliflozin in mice, following repeated oral administration of enavogliflozin (1 mg/kg) for one and two weeks, indicated that kinetics were not significantly different in terms of the Kruskal-Wallis test among three dosage regimes (*p* > 0.05) (Table 7). Based on this statistical similarity, further post-hoc analysis was not performed. Enavogliflozin concentrations and AUC_48 h_ for the kidney, liver, small intestine, and large intestine were not significantly different, regardless of treatment period (single or repeated dose for 7 or 14 days) (Figure 6B–E; Table 7). Likewise, tissue distribution parameters calculated from AUC ratios, following single or repeated oral doses, were not significantly different (*p* > 0.05) (Table 7). 

## 4. Discussion

Previously, we investigated in vitro inhibition mechanisms of SGLT2 and in vivo pharmacokinetic properties of enavogliflozin in mice in a comparison with clinically used SGLT2 inhibitors, dapagliflozin and ipragliflozin. Enavogliflozin, dapagliflozin, and ipragliflozin showed high distribution and long elimination half-lives (t_1/2_) in the kidney and enavogliflozin showed the highest kidney distribution among these three drugs [20]. These properties are thought to be important for efficacy and duration of action of SGLT2 inhibitors [22]. The substrate specificity of enavogliflozin for OAT1 and OAT3 could contribute to renal accumulation [20]. Further, IC_50_ values of enavogliflozin to SGLT2 and SGLT1 are lower than for dapagliflozin and ipragliflozin, suggesting greater affinity to SGLT2 inhibition and, thus. selectively over SGLT1 [20].

In a study for investigating drug metabolism of enavogliflozin in hepatocytes from mouse, rat, dog, monkey, and human, it showed species-different metabolism [23]. Kim et al. identified five phase I metabolites from hepatocytes including, two monohydroxylated metabolites for which CYP3A4 and CYP2C19 were mainly involved, and three dihydroxylated metabolites. Five glucuronide metabolites of enavogliflozin were identified, and UGT1A4 and UGT2B7 were primarily involved in the formation of these glucuronide metabolites [23]. Major phase I metabolites—M1 (6-hydroxy envogliflozin), M3 (subsequent oxidation of M1), and M2 (hydroxylation at the cyclopropyl benzene moiety)—were all found in mouse, rat, dog, monkey, and human hepatocytes. CYP3A4 and CYP2C19 participated in the formation of M1, M2, and M3. UGT1A4, UGT1A9, and UGT2B7 participated in the formation of glucuronide conjugates (U1, U2). These latter metabolites were also found in the hepatocytes of all the species, but the intensity of the respective metabolites were different depending on the species [23]. In vitro hepatic clearance was calculated as 36.5 mL/min/kg for mouse, 8.3 mL/min/kg for rats, 17.9 mL/min/kg for dogs, 18.5 mL/min/kg for monkeys, and 4.5 mL/min/kg for human cells [23]. 

Species differences were also shown in the pharmacokinetics of enavogliflozin in mice and rats. Systemic clearance following intravenous injection of enavogliflosin (0.3–3 mg/kg) in rats was significantly greater than in mice—10.1 ± 1.59 mL/min/kg and 6.35 ± 1.14 mL/min/kg for mice and rats, respectively (*p* < 0.05) (Table 2 and Table 3). However, recovery from the feces following intravenous injection (1 mg/kg) in rats was less than in mice (Table 4 and Table 5). Metabolic activity in rats, thus, appears to be greater than that in mice. However, Kim et al. reported that hepatic metabolic clearance in rats was less than in mice [23]. Collectively, the hepatic and intestinal metabolism may both contribute to enavogliflozin metabolism but intestinal clearance may contribute mainly to systemic clearance, based on higher clearance in rats compared with mice. Oral BA in rats (56.3–62.1%) was also lower than that in mice (84.5–97.2%), which might be explained by favorable intestinal permeability or lower intestinal metabolism in mice. 

Enavogliflozin showed higher biliary excretion compared with renal excretion in both rats and mice following intravenous injection; 39.3% and 15.9% of intravenous injection of enavogliflozin were recovered unchanged from feces samples in mice and rats, respectively (Table 4 and Table 5). In the present study, fecal recovery of unchanged enavogliflozin after oral administration was 51.4% and 45.5% (Table 4). Considering the lower in vitro hepatic clearance in humans than in rats and mice [23], the recovery of enavogliflozin as a parent form following intravenous injection may be greater than that in rats and mice. Moreover, the intestinal first-pass effect of enavogliflozin in humans awaits further investigation to understand the pharmacokinetic and oral BA of enavogliflozin in humans. Conversely, more than 90% of drug-related radioactivity of dapagliflozin and its glucuronide metabolite were recovered in urine from rats and humans after single oral dose administration of [^14^C]dapagliflozin. Thus, renal excretion is the primary route for dapagliflozin and its metabolites [15]. In contrast, cumulative fecal recovery of drug-related radioactivity was 86.9% ± 2.6% following a single oral administration of [^14^C]ipragliflozin. Unchanged ipragliflozin accounted for less than 5% identified in feces and glucuronide metabolite for about 64% identified in feces [16]. Among the fecal recovery (about 85.4–93.7%) of [^14^C]canagliflozin, unchanged canagliflozin, accounted for 3.5–38.7% and hydroxylated metabolites accounted for 7.6–64.3%, following a single oral administration of [^14^C] canagliflozin [24]. 

The steady-state volume of distribution (V_d__,__ss_) of enavogliflozin in mice (3.1–3.2 L/kg) was much higher than total body water (0.7 L/kg), suggesting high extravascular distribution. The drug was highly distributed to the kidney and intestinal tract, displaying a more than 10-fold AUC ratio compared with plasma. Liver distribution was 5.8-fold greater in comparison to plasma AUC (Table 6). Enavogliflozin was not highly distributed to other tissues, such as the heart, lung, brain, spleen, and testis (Table 6). These distribution characteristics were similar for dapagliflozin and ipragliflozin [20]. However, the elimination half-life of enavogliflozin in the kidney was much higher than for dapagliflozin and ipragliflozin [20]. Substrate specificity for renal transporters OAT1 and OAT3 and retained affinity to SGLT2 may contribute to high kidney distribution. The accumulation of enavogliflozin following the repeated oral dosing for 7 and 14 days showed no significant changes in AUC values of enavogliflozin in plasma, kidney, liver, small intestine, and large intestine compared to a single oral dose, suggesting that enavogliflozin can be administered without accumulating in the plasma and major organs within the effective dose range of enavogliflozin (0.1–2 mg in humans) [21]. 

Up to now, little information has been reported regarding the pharmacokinetics and tissue distribution of enavogliflozin. This study aimed to define its pharmacokinetic profiles in mice and rats following intravenous and oral administration, and to investigate the proportionality between doses and plasma exposures. Therefore, we employed non-compartmental analysis to calculate the main pharmacokinetic parameters. However, population PK analysis has been frequently used to guide drug development [25]. In this regards, the pharmacokinetics and kidney distribution results obtained in this study could be used to develop allometric scaling, and to understand the influence of pharmacokinetics on pharmacodynamics along with the in vitro SGLT2 inhibition results and in vivo pharmacological results. 

## 5. Conclusions

This study reports on the linear pharmacokinetic features of enavogliflozin following intravenous and oral administration in a dose range of 0.3–3 mg/kg in both rats and mice. Enavogliflozin was highly accumulated in the kidney, being 41.9-fold higher in kidney AUC than plasma AUC. The drug also displays considerable distribution to the gastrointestinal tract (8.5–12.1-fold plasma AUC ratio) and the liver (5.8-fold plasma AUC ratio). Moreover, the drug shows no accumulation after repeated oral administration in mice. However, species differences in metabolism and oral BA between rats and mice should be recognized as drug development continues. 

## Figures and Tables

**Figure 1 pharmaceutics-14-01210-f001:**
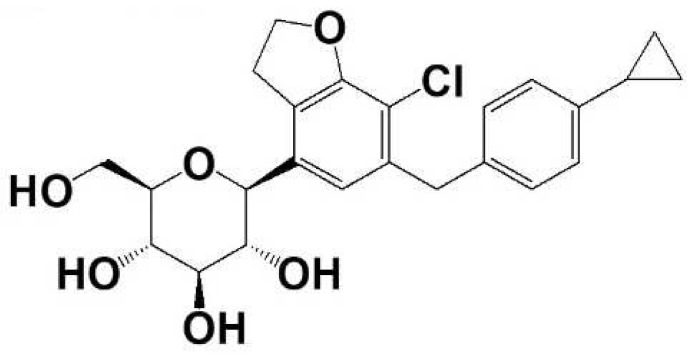
Structure of enavogliflozin.

**Figure 2 pharmaceutics-14-01210-f002:**
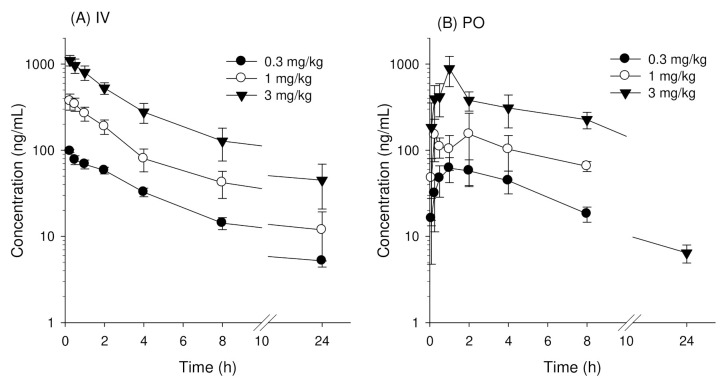
Plasma concentration vs. time profile of enavogliflozin in mice after (**A**) single intravenous (IV) injection of enavogliflozin at doses of 0.3, 1, and 3 mg/kg and (**B**) single oral administration (PO) of enavogliflozin at doses of 0.3, 1, and 3 mg/kg. Data represented as the mean ± standard deviation (n = 6 for each dose).

**Figure 3 pharmaceutics-14-01210-f003:**
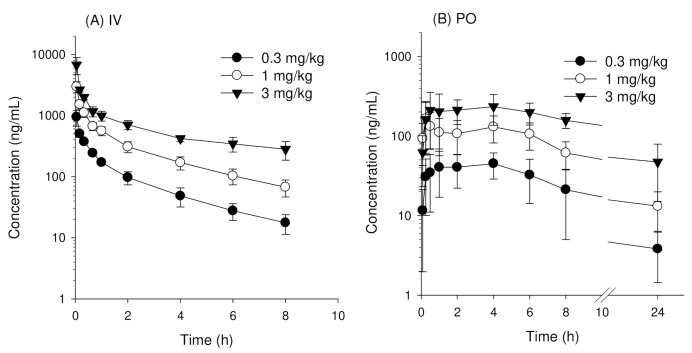
Plasma concentration vs. time profiles of enavogliflozin after (**A**) single intravenous (IV) injection of enavogliflozin at doses of 0.3, 1, and 3 mg/kg and (**B**) single oral administration (PO) of enavogliflozin at doses of 0.3, 1, and 3 mg/kg in SD rats. Data represent mean ± standard deviation (n = 4 per dose for IV, n = 6 per dose for PO).

**Figure 4 pharmaceutics-14-01210-f004:**
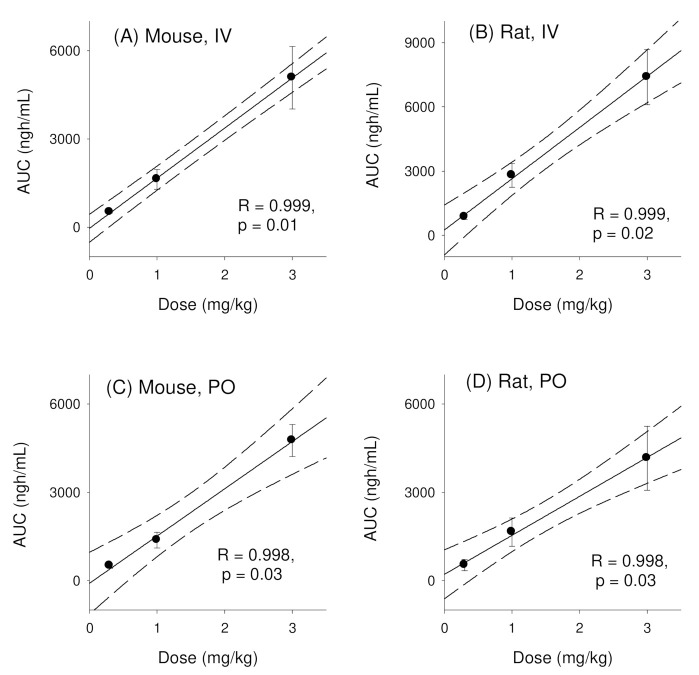
Correlations between AUC or C_0_ values for enavogliflozin and intravenous doses of enavogliflozin in (**A**) mice and (**B**) rats. Correlations between AUC or C_max_ values for enavogliflozin and oral doses of enavogliflozin in (**C**) mice and (**D**) rats. Lines were generated from linear regression analysis and 90% confidence intervals around the geometric mean value. R represents the correlation coefficient and p represents the statistical significance for the regression analysis. Data represent mean ± standard deviation (n = 4 per dose for IV, n = 6 per dose for PO).

**Figure 5 pharmaceutics-14-01210-f005:**
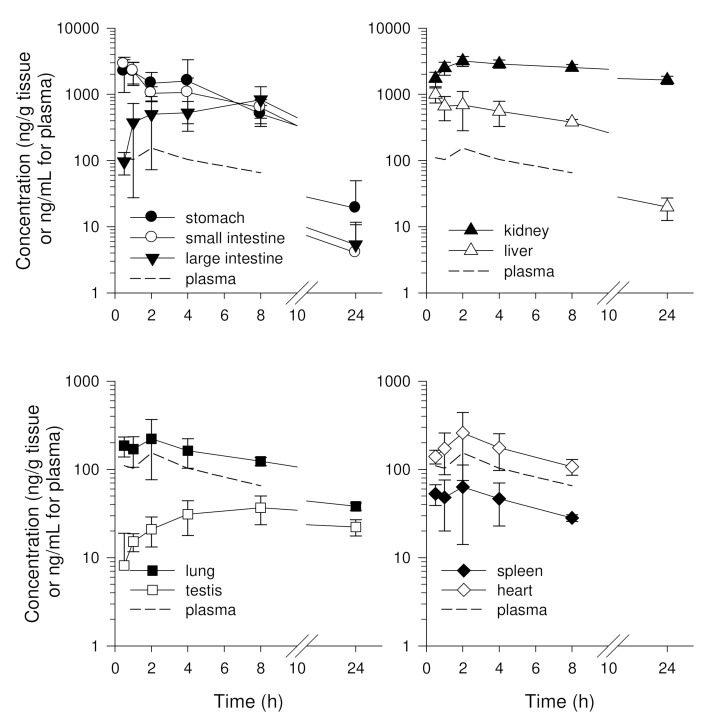
Tissue concentration vs. time profiles of enavogliflozin in stomach, small intestine, large intestine, kidney, liver, lung, testis, heart, and spleen tissues after oral administration of enavogliflozin at a single dose of 1 mg/kg in mice. Plasma concentrations of enavogliflozin are shown as dotted lines. Data are expressed as the mean ± standard deviation (n = 6).

**Figure 6 pharmaceutics-14-01210-f006:**
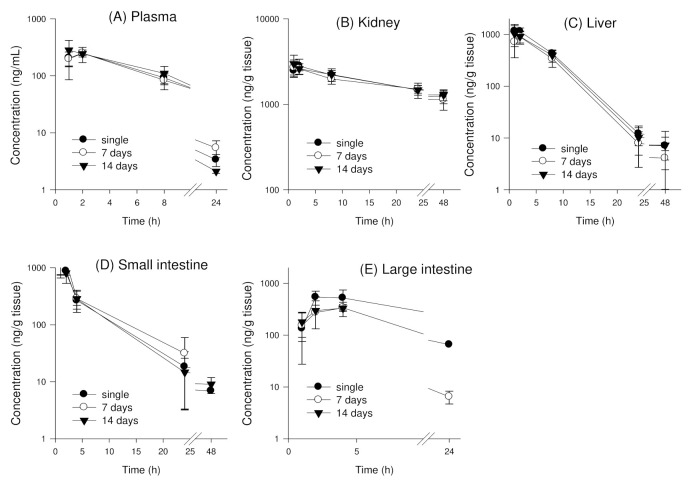
Plasma (**A**) and tissue concentration vs. time profiles of enavogliflozin in mice (**B**) kidney, (**C**) liver, (**D**) small intestine, and (**E**) large intestine following oral administrations of enavogliflozin (1 mg/kg) after a single dose (●, single) or repeated dosing for 7days (○) and 14 days (▼). Data are expressed as mean ± standard deviation (n = 7).

**Table 1 pharmaceutics-14-01210-t001:** Blood sampling procedure in mice.

Treatment	Sampling Time (h)	Group1/4/7 (n = 6)	Group2/5/8 (n = 6)	Group3/6/9 (n = 6)	Sampling (µL)
Intravenous injection (IV, 0.3, 1, 3 mg/kg)	0	RO-right ^1^			80
	0.083		RO-right ^1^		80
	0.25			RO-right ^1^	80
	0.5	RO-left ^2^			80
	1		RO-left ^2^		80
	2			RO-left ^2^	80
	4	AA ^3^			80
	8		AA ^3^		100
	24			AA ^3^	100
**Treatment**	**Sampling Time (h)**	**Group** **10/13/16** **(n =** **6** **)**	**Group** **11/14/17** **(n =** **6** **)**	**Group** **12/15/18** **(n =** **6** **)**	**Sampling** **(** **µL)**
Per oral administration (PO, 0.3, 1, 3 mg/kg)	0	RO-right ^1^			80
	0.083		RO-right ^1^		80
	0.25			RO-right ^1^	80
	0.5	RO-left ^2^			80
	1		RO-left ^2^		80
	2			RO-left ^2^	80
	4	AA ^3^			80
	8		AA ^3^		100
	24			AA ^3^	100

^1^ RO-right: retro-orbital blood sampling—Right eye under anesthesia with isoflurane, ^2^ RO-left: retro-orbital blood sampling–Left eye under anesthesia with isoflurane, ^3^ AA: abdominal artery blood sampling under anesthesia with isoflurane.

**Table 2 pharmaceutics-14-01210-t002:** Dose-independent pharmacokinetic parameters of enavogliflozin after intravenous and oral administrations of enavogliflozin in mice.

Parameter	Dose			
	0.3 mg/kg	1 mg/kg	3 mg/kg	*p* Value
IV administration				
AUC_last_ (ng·h/mL)	478.6 ± 33.6	1512.7 ± 282.3	4657.6 ± 998.0	NC
AUC_∞_ (ng·h/mL)	512.0 ± 30.7	1626.7 ± 335.6	5083.0 ± 1063	NC
AUC_∞_/D	1706.8 ± 102.3	1626.7 ± 335.6	1694.3 ± 354.5	0.810
C_o_ (ng/mL)	125.0 ± 9.4	408.6 ± 103.2	1274.1 ± 186.6	NC
C_o_/D	416.8 ± 31.3	408.6 ± 103.2	424.7 ± 62.2	0.700
CL (mL/min/kg)	9.8 ± 0.6	10.7 ± 2.5	10.2 ± 2.1	0.810
V_d,ss_ (L/kg)	3.1 ± 0.3	3.1 ± 0.6	3.2 ± 0.5	0.755
T_1/2_ (h)	6.0 ± 1.3	6.5 ± 1.4	6.6 ± 0.5	0.484
MRT (h)	5.2 ± 0.4	4.9 ± 0.5	5.2 ± 0.8	0.459
PO administration				
AUC_last_ (ng·h/mL)	487.8 ± 77.9	1363.6 ± 270.0	4735.8 ± 531.9	NC
AUC_∞_ (ng·h/mL)	497.7 ± 79.0	1373.8 ± 269.6	4761.0 ± 543.4	NC
AUC_∞_/D	1658.9 ± 263.5	1373.8 ± 269.6	1587.0 ± 181.1	0.191
C_max_ (ng/mL)	72.1 ± 10.5	216.1 ± 94.1	942.8 ± 230.8	NC
C_max_/D	240.5 ± 35.0	216.1 ± 94.1	314.3 ± 76.9	0.077
T_max_ (h)	1.5 ± 0.5	0.7 ± 0.7	0.9 ± 0.3	0.051
T_1/2_ (h)	4.3 ± 0.5	3.5 ± 0.2	3.3 ± 0.4	0.067
MRT (h)	5.7 ± 0.5	5.7 ± 0.5	5.3 ± 0.5	0.385
BA (%)	97.2	84.5	93.7	-

Absolute bioavailability (BA) was calculated by dividing AUC_∞,_
_PO_ by AUC_∞,_
_IV_; NC: not calculated. Data are expressed as the mean ± standard deviation (n = 6). The p value was calculated among three doses by the Kruskal-Wallis test.

**Table 3 pharmaceutics-14-01210-t003:** Dose-independent pharmacokinetic parameters of enavogliflozin in rats after intravenous and oral administrations of enavogliflozin.

Parameter	Dose			
	0.3 mg/kg	1 mg/kg	3 mg/kg	*p* Value
IV administration				
AUC_last_ (ng·h/mL)	784.5 ± 94.4	2508.8 ± 447.2	5614.1 ± 751.0	NC
AUC_∞_ (ng·h/mL)	849.5 ± 107.7	2798.6 ± 551.4	7387.3 ± 1288	NC
AUC_∞_/D	2831.6 ± 359.0	2798.6 ± 551.4	2462.4 ± 429.3	0.491
C_o_ (ng/mL)	1269.4 ± 516.0	3956.4 ± 1028.2	10,258.1 ± 4094.5	NC
C_o_/D	4231.2 ± 1720.0	3956.4 ± 1028.2	3419.4 ± 1364.8	0.668
CL (mL/min/kg)	6.0 ± 0.8	6.2 ± 1.4	6.9 ± 1.2	0.491
V_d,ss_ (L/kg)	0.9 ± 0.2	1.1 ± 0.1	2.1 ± 0.4	0.053
T_1/2_ (h)	2.6 ± 0.4	2.9 ± 0.4	4.3 ± 0.6	0.021
MRT (h)	2.5 ± 0.4	2.9 ± 0.4	5.1 ± 1.1	0.015
PO administration				
AUC_last_ (ng·h/mL)	489.2 ± 191.9	1348.0 ± 565.2	3254.2 ± 572.5	NC
AUC_∞_ (ng·h/mL)	527.3 ± 193.0	1649.3 ± 483.4	4156.3 ± 1085.3	NC
AUC_∞_/D	1757.6 ± 643.4	1649.3 ± 483.4	1385.4 ± 361.8	0.426
C_max_ (ng/mL)	51.3 ± 15.0	183.0 ± 95.6	300.7 ± 108.2	NC
C_max_/D	171.2 ± 49.9	183.0 ± 95.6	100.2 ± 36.1	0.051
T_max_ (h)	2.5 ± 2.4	2.5 ± 2.5	2.9 ± 2.1	0.690
T_1/2_ (h)	6.3 ± 2.3	6.9 ± 2.1	10.3 ± 7.5	0.625
MRT (h)	8.5 ± 3.2	9.9 ± 3.6	14.9 ± 10.6	0.762
BA (%)	62.1	58.9	56.3	-

Absolute bioavailability (BA) was calculated by dividing AUC_∞,_
_PO_ by AUC_∞,_
_IV_., NC: not calculated. Data are expressed as the mean ± standard deviation (n = 4 for IV, n = 6 for PO). The *p* value was calculated among three doses by the Kruskal-Wallis test.

**Table 4 pharmaceutics-14-01210-t004:** Recovery of enavogliflozin for 72 h in mice.

Dose	Time	Recovery (% of Dose)	
1 mg/kg (IV)	h	Feces	Urine
	0–24	36.3 ± 3.6	6.3 ± 0.8
	24–48	2.7 ± 1.3	0.1 ± 0.0
	48–72	0.4 ± 0.1	ND
	0–72	39.3 ± 3.5	6.6 ± 0.7
1 mg/kg (PO)	h	Feces	Urine
	0–24	50.6 ± 11.1	6.6 ± 0.6
	24–48	0.72 ± 0.4	0.1 ± 0.0
	48–72	0.2 ± 0.1	ND
	0–72	51.4 ± 11.5	6.8 ± 0.6

ND: not detected; Data are expressed as the mean ± standard deviation (n = 3 for IV; n = 4 for PO).

**Table 5 pharmaceutics-14-01210-t005:** Recovery of enavogliflozin for 72 h in rats.

Dose	Time	Recovery (% of Dose)	
1 mg/kg (IV)	h	Feces	Urine
	0–24	15.2 ± 6.2	6.6 ± 0.6
	24–48	0.7 ± 0.8	0.1 ± 0.0
	48–72	ND	ND
	0–72	15.9 ± 5.9	0.7 ± 0.2
1 mg/kg (PO)	h	Feces	Urine
	0–24	44.5 ± 11.8	0.3 ± 0.1
	24–48	0.9 ± 0.4	0.3 ± 0.1
	48–72	0.2 ± 0.0	0.3 ± 0.1
	0–72	50.3 ±8.3	0.3 ± 0.1

ND: not detected; Data are expressed as the mean ± standard deviation (n = 3 for IV; n = 4 for PO).

**Table 6 pharmaceutics-14-01210-t006:** Elimination constant, elimination half-life, and AUC of enavogliflozin in various tissues after single oral doses (1 mg/kg).

Tissue	K (h^−1^)	T_1/2_ (h)	AUC_24 h_ (µg·h/mL for Plasma or µg·h/g Tissue)	AUC Ratio
plasma	0.185	3.7	1.36 ± 0.26	-
kidney	0.024	29	54.5 ± 3.5	41.9 ± 7.7
stomach	0.206	3.4	15.1 ± 6.4	12.1 ± 7.0
small intestine	0.267	2.6	14.2 ± 3.4	10.8 ± 3.0
large intestine	0.204	3.4	11.1 ± 4.3	8.5 ± 4.5
liver	0.159	4.4	8.08 ± 1.1	5.8 ± 0.4
lung	0.072	9.6	3.04 ± 0.29	2.0 ± 0.2
heart	0.194	3.6	2.19 ± 0.35	1.7 ± 0.2
spleen	0.255	2.7	0.58 ± 0.11	0.4 ± 0.0
testis	0.031	22	0.95 ± 0.15	0.5 ± 0.1
brain	NC	NC	NC	NC

NC: not calculated; K: elimination rat constant; T_1/2_: elimination half-life; AUC_24 h_: area under concentration curve for 24 h; AUC ratio was calculated by dividing AUC_24 h, tissue_ by AUC_24 h, plasma.,_ Data are expressed as the mean ± standard deviation (n = 6).

**Table 7 pharmaceutics-14-01210-t007:** AUC values and AUC ratios of enavogliflozin in various tissues after single or repeated oral doses (1 mg/kg) of enavogliflozin in mice.

Tissue	AUC_48 h_ (µg·h/mL for Plasma or µg·h/g Tissue)			*p* Value
	Single Dose	Repeated Dose for 7 Days	Repeated Dose for 14 Days	
Plasma	2.13 ± 0.27	2.09 ± 0.29	2.36 ± 0.69	0.358
Kidney	80.9 ± 5.1	77.2 ± 6.3	81.6 ± 4.8	0.610
Liver	10.2 ± 1.0	7.89 ± 2.1	8.85 ± 1.8	0.213
Small intestine	7.30 ± 0.77	9.39 ± 2.1	7.46 ± 2.3	0.228
Large intestine	5.68 ± 1.6	4.88 ± 1.1	4.90 ± 1.5	0.077
**Tissue**	**AUC Ratio**			* **p** * **Value**
	**Single Dose**	**Repeated Dose for 7 Days**	**Repeated Dose for 14 Days**	
Kidney	38.7 ± 6.3	37.6 ± 5.9	38.4 ± 18	0.444
Liver	4.85 ± 0.64	3.77 ± 0.73	4.37 ± 2.6	0.404
Small intestine	3.47 ± 0.49	4.50 ± 0.89	3.40 ± 1.5	0.251
Large intestine	3.24 ± 1.1	2.33 ± 0.34	2.20 ± 0.71	0.052

AUC_48 h_: area under concentration curve for 48 h; AUC ratio was calculated by dividing AUC_48 h, tissue_ by AUC_48 h, plasma_. Data are expressed as the mean ± standard deviation (n = 7); p value was calculated among three treatment groups by Kruskal-Wallis test.

## Data Availability

Not applicable.

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
