# Peer review of "Pharmacokinetics and Tissue Distribution of Enavogliflozin in Mice and Rats"

_pharmaceutics, 2022, doi:10.3390/pharmaceutics14061210_

Round 1

Reviewer 1 Report

The manuscript submitted by Pang et al. is devoted to pharmacokinetics study of a new drug enavogliflozin in mice and rats following different routes of administration and different doses. The study is very well designed and the conclusions are supported with the obtained results.

I have several minor comments and questions:

1) page 1, line 79: The abbreviation AUC is described not correctly ("area under concentration").

2) Page 6, Fig. 2: The graphs in the panel 2B are difficult to be read. Is it possible to insert an break into the axis X to zoom in the starting points?

Also, to all figures: I recommend to decrease the marker size.

3) Page 7, line 248: The figure 3 is missing. I recommend merging it with the Fig. 2.

4) Page 11, line 331: There are too many graphs in one panel, so that it is impossible to observe any single dependence. I reommend either increasing the scale of the axis Y and make it non-log or separating the graphs iton several panels.

Author Response

  1. Page 1, line 79: The abbreviation AUC is described not correctly ("area under concentration").

[Answer] We corrected our typographical errors as follows:

[Page 2, line 82] The ratio of area under the concentration curve (AUC) ~

  1. Page 6, Fig. 2: The graphs in the panel 2B are difficult to be read. Is it possible to insert a break into the axis X to zoom in the starting points? Also, to all figures: I recommend to decrease the marker size.

[Answer] Thank you for the reviewer’s valuable comments. We modified the graph in Figure 2, 3, and 5 by adding time breaks and decrease the symbol size according to the reviewer’s comments.

3. Page 7, line 248: The figure 3 is missing. I recommend merging it with the Fig. 2.

[Answer] We inserted Figure 3 in the revised manuscript. Since the pharmacokinetic parameter table was located below the figure, we did not merge this figure to Figure 2. We ask the reviewer’s generous understanding on this issue. 

  1. Page 11, line 331: There are too many graphs in one panel, so that it is impossible to observe any single dependence. I recommend either increasing the scale of the axis Y and make it non-log or separating the graphs into several panels.

[Answer] We separated the graph into 4 panels according to the reviewer’s comments to increase the discrimination of the results.   

Reviewer 2 Report

Review comments for pharmaceutics-1730330:

In this manuscript, the authors investigated dose-proportional pharmacokinetics and tissue distribution of enavogliflozin in rats and mice. It was shown that enavogliflozin had a favorable pharmacokinetic profile, significant kidney distribution, and favorable biliary excretion Overall, this work was well-organized and the conclusion was supported by the data present in this work. But there are many problems in this manuscript.

Major comments and revision suggestions are as follows,

  1. In “Pharmacokinetic study” part, why 0.3, 1, and 3 mg/kg were chosen as the dosing concentration of IV and PO administration inrats and mice? Were there previous experimental data reported in the literature?
  2. In the “Recovery of enavogliflozin in mice and rats” part, why are the numbers of experimental animals inconsistent between the two groups (IV:n=3; PO: n=4)? Generally speaking, the number of experimental animals in each group is about 6 to have statistical significance.
  3. Fig 3 is not shown in the manuscript and some forms are not standardized, please check it.
  4. In Fig 5, the unit of plasma sample data is incorrect. It should be “concentration (ng/mL)”.
  5. Page 12, line 352, not Figure 5, but Figure 6. In Figure 6 (D-E), the concentrations of enavogliflozin in the large intestine and small intestine are incorrect units. It should be “ng/g tissue”.

Author Response

  1. In “Pharmacokinetic study” part, why 0.3, 1, and 3 mg/kg were chosen as the dosing concentration of IV and PO administration in rats and mice? Were there previous experimental data reported in the literature?

[Answer] Thank you for the reviewer’s valuable comments. We added the reports regarding the dose selection in the revised manuscript as follows:   

[Page 2, lines 90-96] In a study of comparative pharmacokinetic, pharmacodynamics, and pharmacological effects of various SGLT2 inhibitors in rats, oral dose range of 0.3 ~ 3 mg/kg for ipragliflozin and dapagliflozin and 1 ~ 10 mg/kg for tofogliflozin, canagliflozin, empagliflozin, and luseogliflozin were used as the effective dose range [10]. Considering the effective dose of various SGLT2 inhibitors and IC50 values of enavogliflozin for SGLT2 and SGLT1 [10, 20], we selected 0.3, 1, and 3 mg/kg as oral and intravenous doses of enavogliflozin in this study.

  1. In the “Recovery of enavogliflozin in mice and rats” part, why are the numbers of experimental animals inconsistent between the two groups (IV:n=3; PO: n=4)? Generally speaking, the number of experimental animals in each group is about 6 to have statistical significance.

[Answer] Thank you for the reviewer’s valuable comments. We totally agree with the reviewer’s comments. At first, this study was designed to investigate the major elimination route and the contribution of enavogliflozin parent form to the total recovery with minimum animal number. Therefore, we did not perform the statistical comparison between IV and PO groups and between mice and rats in this manuscript. In addition, the study protocol was already closed. That’s the reason that we could not perform the additional experiments at this time. We will keep this in mind for the next experiments and ask the reviewer’s generous understanding on this issue.

  1. Fig 3 is not shown in the manuscript and some forms are not standardized, please check it.

[Answer] Thank you for the reviewer’s valuable comments. We added Figure 3 during the revision.

  1. In Fig 5, the unit of plasma sample data is incorrect. It should be “concentration (ng/mL)”.

[Answer] As the reviewer suggested, we corrected our mistakes in the revised manuscript:

  1. Page 12, line 352, not Figure 5, but Figure 6. In Figure 6 (D-E), the concentrations of enavogliflozin in the large intestine and small intestine are incorrect units. It should be “ng/g tissue”.

[Answer] Thank you for the reviewer’s valuable comments. We corrected our mistakes in the revised manuscript.

Reviewer 3 Report

The aim of this study was to investigate the pharmacokinetics of enavogliflozin in mice and rats. Enavogliflozin is a novel sodium-glucose cotransporter-2 inhibitor, and for this reason, this study is of interest to readers, 

Some points need to be reconsidered by the authors: 

Major comments:

1. Tables 2 and 3 in conjunction with lines 207-209:
Before using the two-independent t-test or the one-tailed ANOVA, make sure that the variables follow a normal distribution. Since your sample size is small, the Shapiro-Wilk method is the method of choice in this case.
If the normality test did not yield a normal distribution, you should use the non-parametric Mann-Whitney test.

2. Table 7.
For the purposes of repeated dosing, the methodology described in lines 207-209 is not correct.
In this case, perform the following two steps:
(a) normality test (as described in comment 1 above).
b) pairwise comparison, e.g., repeated measures t-test ANOVA (if a normal distribution is found) or Friedman test (for non-parametric methods). 

3. The methodology section should be completed by the description of the statistical methods mentioned above.

4.The pharmacokinetic analysis was based on simple non-compartmental methods.
It is recommended that you use population pharmacokinetics to obtain more robust estimates of pharmacokinetic parameters. Since your sample size is small (i.e., 6 mice/rats per group), the population pharmacokinetic method is by far more appropriate. 

Minor comments:
1. Lines 207-209: please rephrase the expressions "...to analyze three results ...", "... two results."
There are no "results" to analyze. It is better to use the term "group of ..." or "groups". 

Reviewer 4 Report

I think this is a well-designed research.

Sorry if I missed it, but please list the Enavogliflozin lot number.

Author Response

I think this is a well-designed research. Sorry if I missed it, but please list the Enavogliflozin lot number.

[Answer] Thank you for the reviewer’s comment. We added the Lot No. for enavogliflozin: Enavogliflozin (Lot No. E2016-085-27-2) (Figure 1)

Round 2

Reviewer 3 Report

The authors responded to the previous comments by revising the "Methods" section and stating that data does follow normal distribution in all cases, and thus the previously applied parametric methods were valid.

- Comment 1:

However, no results from the normality testing were added to the manuscript. 

Since this point is very crucial for the validity of the statistical analysis and the conclusions drawn from this study, I would suggest:

1. Include at least the p-values derived from the Shapiro-Wilk test in the text.

2. Add the results of all normality testing in the form of Tables and QQ plots in the Appendix.

- Comment 2:

It is widely known that pharmacokinetic parameters do not follow Normal distribution and for this reason, regulatory authorities suggest the application of log-normal transformation before applying parametric methods. This is fully contrasted with the comments you made that, in all cases, you found normal distribution.

How can you justify this finding ?

Round 3

Reviewer 3 Report

Issues resolved